# Quantifying the relationship between SARS-CoV-2 viral load and infectiousness

**Aurélien Marc[1]\*, Marion Kerioui[1], François Blanquart[1,2], Julie Bertrand[1], Oriol Mitjà[3,4], Marc Corbacho-Monné[3,5,6], Michael Marks[7,8,9], Jeremie Guedj[1]\***

[1]Université de Paris, IAME, INSERM, Paris, France; [2]Centre for Interdisciplinary Research in Biology (CIRB), Collège de France, CNRS, INSERM, PSL Research University, Paris, France; [3]Fight AIDS and Infectious Diseases Foundation, Hospital Universitari Germans Trias i Pujol, Badalona, Spain; [4]Lihir Medical Centre, International SOS, Londolovit, Papua New Guinea; [5]Hospital Universitari Parc Taulí, Sabadell, Spain; [6]Facultat de Medicina–Universitat de Barcelona, Barcelona, Spain; [7]London School of Hygiene and Tropical Medicine, London, United Kingdom; [8]Hospital for Tropical Diseases, London, United Kingdom; [9]Division of infection and Immunity, University College London, London, United Kingdom

**Abstract** The relationship between SARS-CoV-2 viral load and infectiousness is poorly known. Using data from a cohort of cases and high-risk contacts, we reconstructed viral load at the time of contact and inferred the probability of infection. The effect of viral load was larger in household contacts than in non-household contacts, with a transmission probability as large as 48% when the viral load was greater than $10^{10}$ copies per mL. The transmission probability peaked at symptom onset, with a mean probability of transmission of 29%, with large individual variations. The model also projects the effects of variants on disease transmission. Based on the current knowledge that viral load is increased by two- to eightfold with variants of concern and assuming no changes in the pattern of contacts across variants, the model predicts that larger viral load levels could lead to a relative increase in the probability of transmission of 24% to 58% in household contacts, and of 15% to 39% in non-household contacts.

**\*For correspondence:**
aurelien.marc@inserm.fr (AM);
jeremie.guedj@inserm.fr (JG)

**Competing interests:** The authors declare that no competing interests exist.

## Introduction

After more than 18 months of an unprecedented pandemic, some key aspects of virus transmission remain poorly understood. While respiratory droplets and aerosols have rapidly been demonstrated as a major route of transmission of SARS-CoV-2 (*Tang et al., 2020*), the role of the viral load as a driver of infectiousness has been established (*He et al., 2020*) but not quantified. This lack of evidence is due to the fact that high-risk contacts occur mostly before the index has been diagnosed, with no information on the viral load level at the time of the contact. The relationship between viral load and infectiousness determines the timing of transmission, the inter-individual heterogeneity in transmission, and ultimately the impact of interventions (contact, case isolation, vaccination) on transmission. In the context of variants of concern, that are associated with larger viral loads (*Teyssou et al., 2021*; *Liu et al., 2021*; *Elie et al., 2021*; *Cosentino et al., 2021*; *Jones et al., 2021*), it becomes even more critical to delineate the contribution of viral load from other factors associated with an increased transmission. Further, as antiviral drugs and vaccine strategies are being implemented, that dramatically reduce the amount of viral shedding (*Levine-Tiefenbrun et al., 2021*), it is essential to understand how they may contribute to a reduction in virus transmission.

One of the most documented clinical study to address the question of viral load and infectiousness has been obtained through individuals included in a randomised controlled trial conducted in March-April 2020 in Spain, that aimed to assess the efficacy of hydroxychloroquine on SARS-CoV-2 transmission (*Mitjà et al., 2021*; *Marks et al., 2021*). Overall, 282 index and their 753 high-risk contacts were frequently monitored to assess their virological and clinical evolution. An association was found between the probability of being infected after a high-risk contact and the viral load measured at the time of diagnosis in the index case (*Marks et al., 2021*). This suggests that viral load is associated with transmission; however, it does not quantify the role of viral load in disease transmission, as the viral load at the exact time of the contact remains unknown and may greatly differ from that measured, several days later, at the time of diagnosis.

In order to study in detail the role of viral load on the probability of transmission, we reanalysed these data by using a within-host model of viral dynamics (*Néant et al., 2021*; *Gonçalves et al., 2020*) to reconstruct the viral load levels of the index cases at the time of contact, and to infer the relationship between viral load and the probability of transmission after a high-risk contact. Further, we used the model to predict the effects of changes in viral load levels on the probability of transmission, representing the effects of infection with a variant of concern or infection in an individual in which vaccine would confer a partial protection against viral replication.

## Results

### Baseline characteristics

A total of 259 index cases and their 582 high-risk contacts (simply called contacts in the following) were included in our analysis (*Figure 1—figure supplement 1*).

The majority of index cases were female (72%) with a median age of 42 (90% Inter Quantile Range (IQR): [24, 61]). A total of 544 swab samples were performed at days 0, 3 and 7 days after study inclusion. The first swab was performed after a median time of 4 days (90% IQR: [2, 6]) after symptom onset. The maximum median viral load obtained during follow-up was 8.4 $\log_{10}$ copies per mL (90% IQR:[5.1, 10.6]).

The majority of contacts were female (56%) with a median age of 41 (90% IQR: [20, 65]). The form of contacts was categorized as either household (60%) or non-household (40%).

Overall, 87 household contact led to an infection (proportion of transmission of 24.9%) and 29 non-household contacts led to an infection (proportion of transmission of 12.4%). The majority of contacts (65%) and of infection events (65%) occurred ±1 day from symptom onset of the index cases (*Figure 1—figure supplement 2*).

### Viral dynamic model

We used a target cell limited model to reconstruct the viral load kinetics of the index cases over time, assuming that the incubation period has a log-normal distribution with a mean value of 5 days (*Néant et al., 2021*; *Lauer et al., 2020*). Although several models relating viral load to infectiousness were evaluated (see below), they all provided nearly identical fits to the viral load data predicted in the index cases (*Figure 1*). Additionally, we tested several models with a fixed incubation period ranging from 4 to 7 days, and they all yielded similar results (*Supplementary file 1*). In the best model (Model M2, see below), the basic within-host reproductive number, $R_0$, quantifying the number of cell infections that occur from a single infected cell at the beginning, was estimated at 13.6, the loss rate of productively infected cells, $\delta$, at 0.84 d$^{-1}$ (corresponding to a half-life of 20 hr) and viral production $p$, at $2.8 \times 10^5 \text{cells}^{-1}$ d$^{-1}$ (*Table 1*). When reconstructing the viral load profiles, the model predicted that the median peak viral load coincided with symptom onset, with a median peak value of 9.4 $\log_{10}$ copies per mL (90% IQR: [8.0, 10.0]).

We tested several models of probability transmission (see Materials and methods) and estimated the parameters of both viral dynamics and probability of transmission simultaneously. The two model assuming an effect of viral load on the probability of transmission (Model M2 and M3) provided an improvement in BIC as compared to the model M1, supporting an effect of viral load on the probability of infection. In both models, viral load was significantly associated with the probability of transmission after household contact (p<0.01, Wald test on $\gamma_1$); however, the effect was lower after non-household transmission (p<0.05, Wald test on $\gamma_2$). Because we fixed the probability of transmission

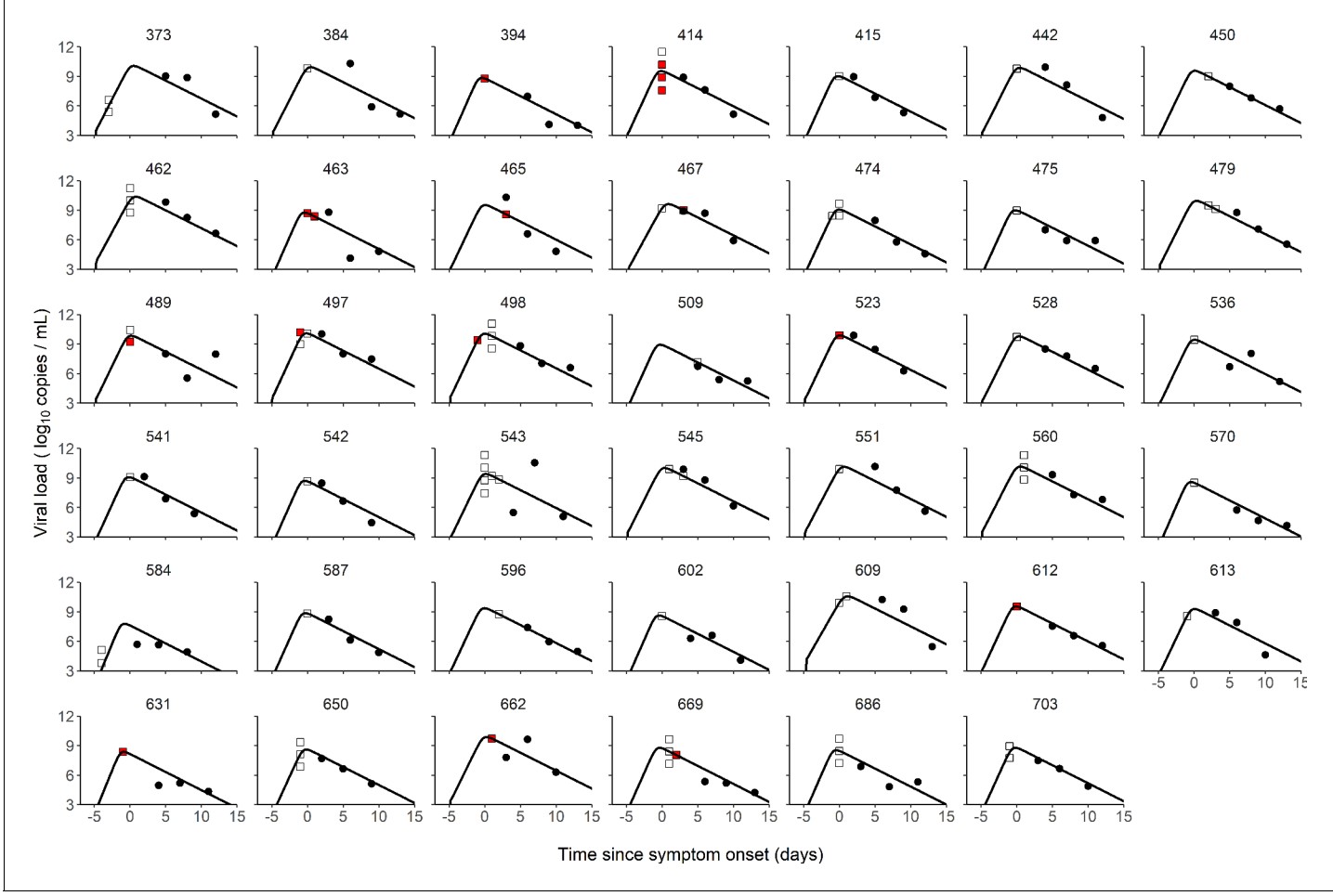

**Figure 1.** Individual fits of viral dynamics in index cases and occurrence of high-risk contacts. Black dots represent the measured viral load. Squares indicate documented high-risk contacts, with empty squares representing contacts without transmission, and red squares representing contacts with a subsequent infection. Results obtained in the 41 index cases having three viral load measurements.

The online version of this article includes the following figure supplement(s) for figure 1:

**Figure supplement 1.** Flow chart of data selection.

**Figure supplement 2.** Distribution of contacts.

to 5% for viral load levels below six $\log_{10}$ copies/mL, which is generally the threshold for virus culture in vitro (*Jones et al., 2021*; *Néant et al., 2021*; *van Kampen et al., 2021*; *Mollan et al., 2021*), we tested models with threshold values ranging from 4 to 8 $\log_{10}$ copies/mL and they all yielded similar results (*Supplementary file 2*).

As a mean to evaluate the model adjustment to data, we also used simulations to compare the observed proportion of transmission in the original data to the mean probability of transmission obtained from the simulated individuals. The model-based simulations showed good agreement with the observed data, and reproduced well the increase in the transmission probability associated with higher viral load level (*Figure 2*). The model predicted that the mean probability of transmission increased from the fixed nominal value of 5% for viral load levels < 6 $\log_{10}$ copies per mL, to as much as 48% and 20% for viral load $\geq$ 10 $\log_{10}$ copies per mL for household and non-household contacts, respectively. This is close to the values of 56% and 20% obtained on the predicted individuals. (*Figure 2*).

The model considers two levels of individual variability, one on the viral load dynamics (*Chen et al., 2021*) (as measured by the standard deviation of the associated random effects, $\omega_{R_0}, \omega_\delta$ and $\omega_p$), and another one on the probability of transmission (with a standard deviation $\omega_\beta$). Of note, $\omega_\beta$ was equal to 85%, indicating that several other factors are involved in the transmission

**Table 1.** Parameters estimates of the three candidate models.

$R_0$, within-host basic reproductive number; $\delta$, loss rate of infected cells; $p$, rate of viral production; $\gamma_1$ represents the effect of household contacts on the transmission probability; $\gamma_0$ represents the effect of non-household contacts on the transmission probability. M1 assumes that transmission probability does not depend on the viral load. M2 and M3 use different parametric functions to relate the transmission probability to viral load at the time of contact. The distribution of the incubation period was fixed to values from the literature (see Materials and methods).

| | Parameter estimates (RSE %) | | | | | |
| | No effect of viral load (M1) | | Logit-Linear (M2) | | Log-Linear model (M3) | |
| | Fixed effect | Random effect SD | Fixed effect | Random effect SD | Fixed effect | Random effect SD |
| --- | --- | --- | --- | --- | --- | --- |
| Incubation period ($d$) | 5 | *0.125* | 5 | *0.125* | 5 | *0.125* |
| $R_0$ | 12.20 (14) | 0.32 (34) | 13.60 (15) | 0.38 (21) | 13.40 (22) | 0.423 (35) |
| $\delta(d^{-1})$ | 0.83 (1) | 0.019 (47) | 0.84 (4) | 0.037 (77) | 0.832 (100) | 0.023 (74) |
| $p$ ($cells^{-1}.d^{-1}$) | $1.97 \times 10^5$ (41) | 2.38 (9) | $2.8 \times 10^5$ (50) | 2.35 (8) | $2.40 \times 10^5$ (47) | 2.3 (9) |
| $\gamma_1$ | 1.28 (38) | 0.82 (55) | 0.49 (20) | 0.85 (32) | 0.47 (6) | 0.545 (23) |
| $\gamma_2$ | 0.57 (62) | | 0.21 (44) | | 0.25 (17) | |
| BIC | 2502 | | 2497 | | 2500 | |

probability, even after adjustment for viral load levels (see *Supplementary file 3* for the results obtained with a model assuming a similar value for $\beta$ in all individuals). This variability is shown on *Figure 3*, where 1000 individuals were sampled in the population distribution to obtain the probability of transmission over time and across individuals. Over the time of infection, the median probability of transmission peaked at the time of symptom onset with a mean value of 29% in household contacts. However, there was large inter-individual variabilities due to both viral load levels and individual characteristics, with a 90% inter quantile range of 6-96% (*Figure 3*). The peak of transmission was much lower in non-household contacts, with a mean value of 13% (90% IQR: [5, 38]). As a consequence of our assumption that the probability of transmission after a high-risk contact returned to baseline level when viral load dropped below 6 $\log_{10}$ copies per mL, the window for infection was shorter than the duration of viral shedding. The probability of transmission was above 5% for a median duration of 12 days (90% IQR: [9, 15]).

## Sampling the generation interval

As a mean to validate the model prediction, we also calculated the generation interval, that is the time elapsed between the infection of an individual and the infection of a contact. We considered two potential distributions of contact times, one in which the rate of contacts is constant during the whole considered period, and one in which the rate of contacts decreases rapidly after 5 days, reflecting self-isolation and/or diagnosis (*Figure 4A*). The median generation interval was estimated to be 5.1 days (90% IQR: [1, 10]) and 4.8 days (90% IQR: [1, 11]) for household and non-household contacts respectively, when a time-varying rate of contacts was used. Those estimates are close, albeit with higher variability, to what has been found in other studies (*Cereda et al., 2020*; *Bi et al., 2020*). When using a constant contact rate, we obtained larger estimates of 7.7 days (90% IQR: [2.4, 17]) and 8.2 (90% IQR: [1.6, 18]) in household contacts and non-household contacts, respectively (*Figure 4B*). Because the time varying contact rate was more realistic (*Cereda et al., 2020*; *Bi et al., 2020*; *Ferretti et al., 2020*; *Wu et al., 2020*), we used it as our central scenario in what follows.

## Impact of variants of concern and vaccination on the probability of transmission

We used the model to characterize the effects of changes of viral load dynamics due to infection with variants of concern. For that purpose, we evaluated the impact of a change in the viral production rate, $p$, by a fold 2–100, which corresponds to an average increase in viral load of 1–7 cycle thresholds (Ct), at each time point (*Figure 4—figure supplement 1*). As a metrics of comparison, we

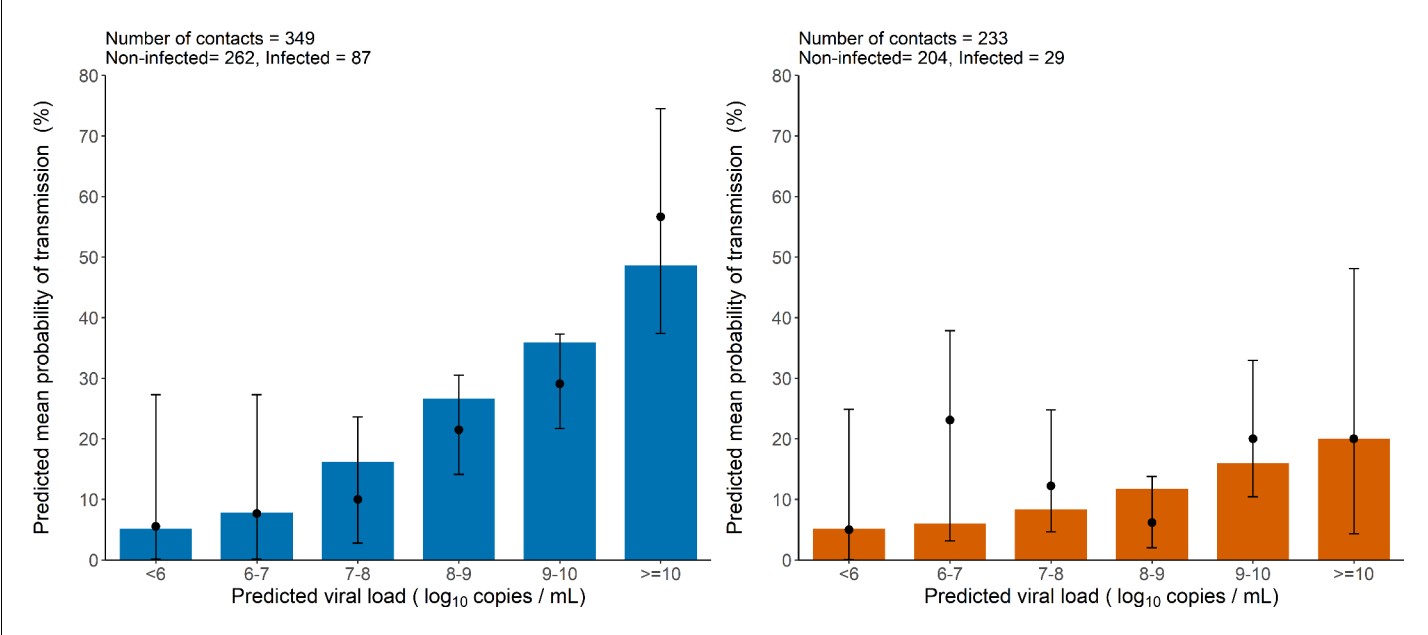

**Figure 2.** Model-based predictions of the effect of viral load on the risk of transmission and comparison to observed data. Bars represent the mean predicted probability of transmission obtained from 1000 simulations of the model M2 and stratified by viral load level at the time of contact. Black dots are the proportion of transmission events observed in the data stratified by the predicted viral load of the index cases at the time of contact (along with their 95% confidence intervals). Household contacts (Left). Non-household contacts (Right).

calculated in each scenario the average probability of transmission after a high-risk contact in the 20 days following infection (see methods).

For the baseline scenario using the parameters estimated in our population, the average transmission probability was 18% and 9% for household and non-household contacts, respectively.

With an increased value of viral production, p, by a factor 2, which corresponds to the viral load increase caused by B1.1.7 strain in large-scale epidemiological studies (*Golubchik et al., 2021*; *Roquebert et al., 2021*; *Kidd et al., 2021*), the average probability of transmission would increase to 22% and 11% for household and non-household contacts respectively. With a fourfold increase, as suggested elsewhere (*Teyssou et al., 2021*), the average probability of transmission would increase to 26% and 12% for household and non-household contacts, respectively (*Figure 4C*). The estimates for the P1 and B1.1.351 variants are less established, with values ranging from a twofold (*Teyssou et al., 2021*) to a 10-fold increase (*Naveca et al., 2021*). Assuming an increase by eightfold of the viral load, the average probability of transmission would increase to 29% and 13% for household and non-household contacts, respectively. As compared to the results observed with the historical virus a two-, four-, or eightfold increase in viral production rate would therefore lead to a relative increase in the average transmission probability of 24, 42, or 58% for household contacts, and of 15, 27, or 39% for non-household contacts (*Figure 4D*). Because increasing the production rate mostly impacts the early viral dynamics and less the post-peak dynamics (*Figure 4—figure supplement 1*), the effects of VOC is lower when a uniform distribution is used. In other words, self-isolation after symptoms or a positive test implies that more transmission happens early in infection, thus amplifies the impact of the viral production rate on transmission. In this case, we estimated a relative increase in the average transmission probability of 6, 15, or 24% for household contacts, and of 4, 10, or 16% for non-household contacts.

Conversely, we studied the effects of lower levels of viral load, as expected from a partial protection conferred by vaccination. Epidemiological studies in Israel reported a 3-5-fold lower viral load in infected vaccinated individuals as compared to unvaccinated individuals (*Levine-Tiefenbrun et al., 2021*). Assuming a reduction by a factor 4 of the viral production rate, *p*, would lead to an average probability of transmission of 7% and 6% for household and non-household contacts respectively (*Figure 4*). In other studies relying on systematic repeated viral testing in both symptomatic and

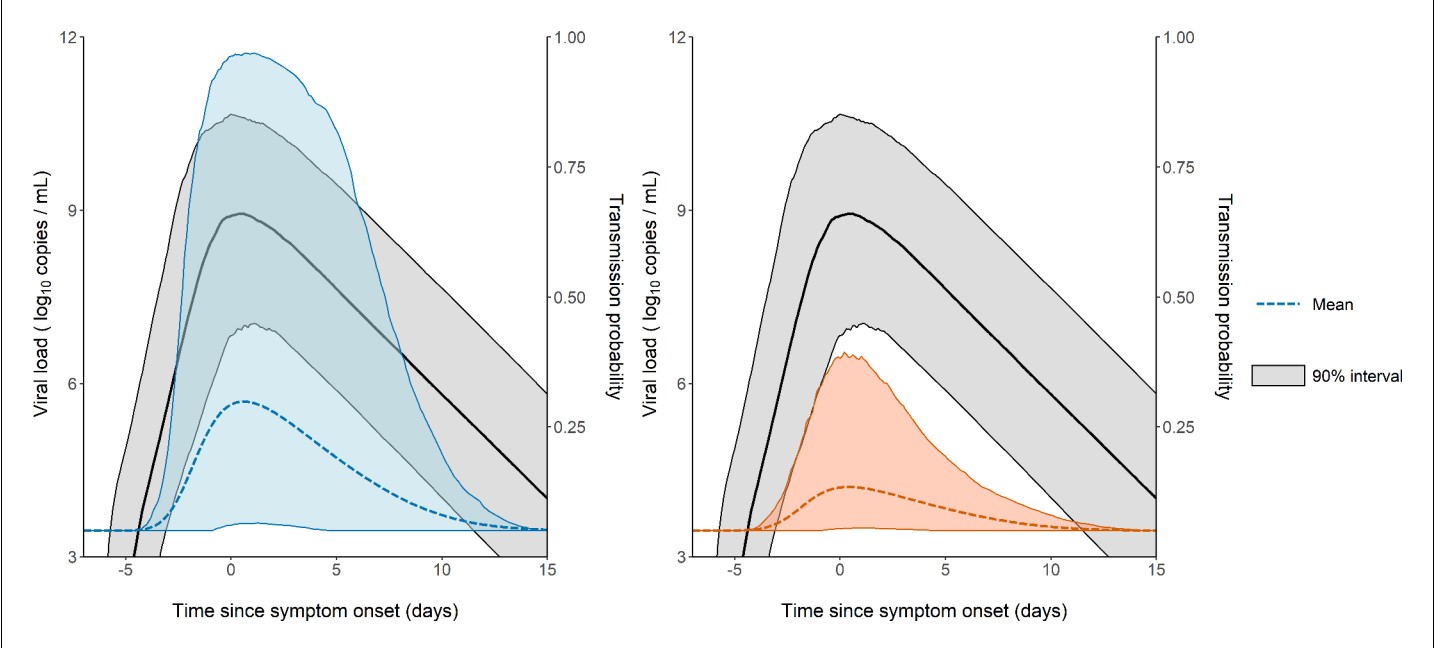

**Figure 3.** Model-based predictions of the dynamics of viral load and infectiousness over time. Prediction interval of the viral load (black) and the probability of transmission over time after a high-risk contact obtained from 1000 simulations of the model. The shaded area represents the 90% inter quantile range. Household contacts (Left). Non-household contacts (Right).

asymptomatic individuals, the effect of vaccine was much more dramatic, with a 30-100-fold reduction in viral load levels (*McEllistrem et al., 2021*; *Thompson et al., 2021*; *Bailly et al., 2021*). Assuming a reduction of 16-fold (~4 Ct) of the viral load, the average probability of transmission would decrease to its nominal value of 5% for both household and non-household contacts. As compared to the results observed with the historical virus, a 4- or 16-fold reduction in viral production rate would lead to a relative decrease in the average transmission probability of 61% or 72% for household contacts and of 38 or 47% for non-household contacts. The effect of vaccination is lower if a uniform distribution of contact is used, with a relative decrease in the average transmission probability of 23 or 66% for household-contacts and of 11 or 38% for non-household contacts (*Figure 4*).

Results obtained with model M3 were largely consistent and are given in *Figure 4—figure supplement 2*.

## Discussion

Here, we quantified the impact of viral load on infectiousness using data obtained in a prospective cohorts of index and contact cases (*Mitjà et al., 2021*). The effect of viral load was particularly large in household contacts, with a mean transmission probability that increased to as much as 48% when the viral load was over 10 $\log_{10}$ copies per mL. Consistent with reports suggesting that the probability of transmission (*Edwards et al., 2021*) greatly vary between individuals, the effect of viral load was individual-dependent. For instance, at the peak of infectiousness, the mean probability of transmission during household contact was 29% with a 90% inter quantile range of 6–96%.

The model also provided information on the effects of variants on disease transmission. We relied on results found in both large-scale epidemiological data and longitudinal evaluation of Ct values (*Elie et al., 2021*; *Cosentino et al., 2021*), that reported an average increase of the B1.1.7 virus by 1-2 Ct (*Teyssou et al., 2021*; *Golubchik et al., 2021*; *Roquebert et al., 2021*), which can be reproduced in our model by assuming that viral production increases by a factor 2-4. Alternatively, as only the product $p \times T_0$ can be identified, this could also be due to B1.1.7 being able to infect twice as much target cells, as suggested by the fact that the N501Y substitution improved the affinity of the viral spike protein (*Liu et al., 2021*). Regardless of the origin of this increased viral load, we estimated that an increase of viral load by a factor of 2, 4, or 8 would lead to a relative increase in the

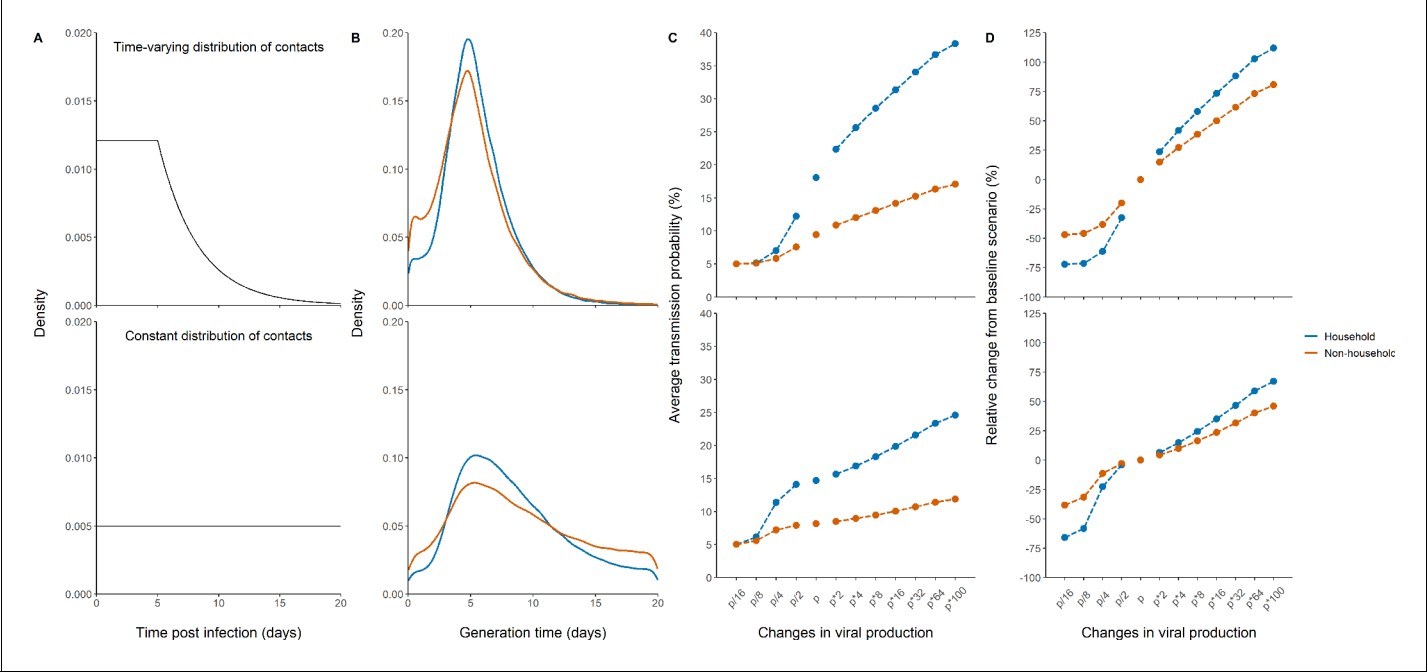

**Figure 4.** Effects of variants of concern and vaccination on transmission for different distributions of contacts. (**A**) We considered a rate of contacts that could either decline after 5 days (top) or remain constant for the whole considered period (bottom). (**B**) Distribution of the generation interval using model M2 under each scenario. (**C**) Impact of changes in viral production on the average probability of transmission. (**D**) Impact of changes in viral production on the relative change from the baseline scenario in model M2.

The online version of this article includes the following figure supplement(s) for figure 4:

**Figure supplement 1.** Effect of changes in the viral production on the viral load and transmission probabilities trajectories.

**Figure supplement 2.** Impact of changes in the viral production rate on the average probability of transmission with model M3.

average transmission probability of 24, 42, or 58% in household contacts and of 15, 27, or 39% for non-household contact. As raised by one of the reviewers, it is important to recognize that the association between VOC and viral load levels relies on observational studies, with data mostly collected after symptom onset, both factors limiting a formal causation. In fact, another modelling study performed in a small population of frequently sampled individuals diagnosed early in their infection did not find an effect of B1.1.7 on viral kinetics (**Ke et al., 2021**).

Conversely, vaccination rollout is expected to confer a large level of protection, partly due to lower virus carriage in infected individuals. The exact magnitude of this decrease is difficult to quantify, and depends on the design of the study that relied on systematic testing or included only symptomatic individuals. This may explain the variability in the reports from the literature from 5- to 100-fold reduction in viral load levels (**McEllistrem et al., 2021**). Whatever the exact value, our predictions indicate that reductions of fourfold or more will dramatically reduce the probability of transmission carried by vaccinated infected individuals.

Our study has important limitations. First, the reporting of high-risk contacts is prone to several biases. One of them is the fact that at the time where the study was conducted, the role of pre-symptomatic transmission was not known. This could explain why a large number of high-risk household contact were reported to occur the day of symptom onset (**Figure 1—figure supplement 2**). Also, it is possible that recollection bias leads to an overestimation of contacts reporting on the day of symptom onset. Because this will equally affect contacts that resulted in a transmission event and those that did not lead to a transmission event, it is unlikely that our estimates of transmission will be affected by this bias. To address a potential overestimation of the contacts occurring at symptom onset, we used two theoretical distributions of contacts in our simulations, assuming either a constant distribution of contact during the infectious period, or a more realistic scenario in which most contacts occurred during the first 5 days after infection. Also, we assumed the same patterns of contacts in our different scenarios. Although there are no data on these aspects yet, it is possible that

larger levels of viral shedding could lead to a more severe infection or, inversely, that lower viral load could produce milder infections, thereby modifying the incubation period and more generally the patterns of contact. Another important limitation is that household contacts may not be unique and could occur multiple times. Because we had no information on these contacts, we did not conduct specific analyses on repeated contacts, but this is something that future epidemiological studies will need to investigate. Finally, it is always possible that infection observed in contacts individuals did not originate from the identified index case. In most infected contacts, we did not have data on the time of symptom onset, making it difficult to detect unplausible transmission event. However, the temporality of symptoms would not be sufficient to bring a decisive information on the infection event. Indeed, the study was conducted during the first epidemic wave in Spain, where most individuals, including in hospital settings, had not yet applied social distancing and masking, causing dozens of thousands of individuals infected every day. Both the possibility of repeated contacts in household and infection of contacts outside the identified contact network may have led us to overestimate the difference in the probability of transmission between household and non-household contacts. Specifically, infections outside of the identified probability contact would flatten the estimated relationship between viral load and transmission compared to the true relationship. It is also important to note that viral load data in index cases were collected on average 3–4 days after symptom onset, in the declining phase of viral load, several days after most of the contacts had occurred. Although our population parameters were estimated with a reasonable precision (*Table 1*), it nonetheless brings uncertainty on the predictions of individual trajectories. This limitation is inherent to the nature of SARS-CoV-2, where the peak viral load coincides with symptom onset, making difficult to obtain data during the replicating phase of the virus where individuals are largely asymptomatic.

Beside viral load, several factors are associated with a transmission event. One important one is face masking, for both the index and the contact. In the original analysis of *Marks et al., 2021*, the use of face mask by contacts was not found associated with a decreased viral load, but this probably reflects the lack of more detailed data on the type of mask, the use of other personal protective equipment and infection control practices. It is also important to recall that face masking was poorly reported and was missing in about 35% of contacts, limiting statistical power (*Supplementary file 4*). The use of face mask by index cases was not collected in the original study. This information might be of a greater importance as it has a far more substantial effect on viral shedding and thus on transmission. Collecting this information in future studies should probably contribute to a reduction in the variance of the random effect parameter associated with transmission ($\omega_\beta$).

To conclude, our study quantifies the probability of infection according to viral load level after a high-risk contact. This relationship can be used to predict the effects of changes in virus paradigm, caused by the emergence of new variants and/or the rollout of vaccination. We estimate that two- to eightfold increase in viral load level observed with variants of concern could lead to an increase in the probability of transmission by 24–58% in household contacts.

## Materials and methods

### Data collection

Data used come from a cluster-randomised trial which included individuals with PCR-confirmed COVID-19 and their close contacts, and evaluated the efficacy of hydroxychloroquine as a pre- or post-exposure prophylaxis. The trial was conducted between March, 17 and April 28, 2020 in three out of nine health-care area in Catalonia, Spain. More details on the study protocol and main results can be found in the original publication (*Mitjà et al., 2021*).

### Study participants

All index cases were individuals aged 18 years or older, identified by the Catalan epidemiological surveillance system, with no hospitalisation, nasopharyngeal PCR positive results at baseline and mild symptoms onset within 5 days of inclusion and had no reported symptoms of SARS-CoV-2 infections in their accommodation or workplace within the 14 days before enrolment. High-risk contacts were adults with a recent history of exposure (i.e. >15 min within 2 m up to 7 days before enrolment) and absence of COVID-19 like symptoms within the 14 days preceding enrolment, and who had an increased risk of infection (e.g. health care worker a household contact, a nursing-home worker, or a

nursing-home resident). Contacts were quarantined upon enrolment to the study. In the original study, 282 index cases and the resulting 753 contacts were enrolled (*Marks et al., 2021*); here we did not include three index individuals (and their corresponding 25 contacts) for which no viral load data was available, eight index individuals (19 contacts) for which no viral load was detected at any time point, and 12 index cases (127 contacts) for which no date of contact was available. Thus, our analysis was performed on 259 index and 582 contacts (*Figure 1—figure supplement 1*). In 12 index cases, the date of symptoms onset was not known and was imputed to 4 days before their first swab sampling, which corresponds to the median value observed in the population study. Type of contact was considered as household or non-household, the latter included nursing home contacts, health-care worker and other undefined contacts.

## Reconstructing viral load in index cases using a viral kinetic model

We used a target cell-limited model to reconstruct nasopharyngeal viral kinetics in index cases (*Néant et al., 2021*; *Madelain et al., 2018*; *Baccam et al., 2006*). The model includes three populations of cells, namely Target cells ($T$), infected cells in their eclipse phase ($I_1$) and productively infected cells ($I_2$). Target cells ($T$) are infected at a constant rate $\beta$ by infectious virus ($V_I$). Infected cells enter an eclipse phase at a rate $k$ before becoming productively infected cells ($I_2$). We assumed productively infected cells have a constant loss rate $\delta$. Virions are released from productively infected cells at a rate $p$ and are loss at a rate $c$. A proportion $\mu$ of produced viruses are infectious ($V_I$) and the remaining ($1 - \mu$) are non-infectious viruses ($V_{NI}$), both are cleared at a rate $c$. The model can be written as follows:

$$\frac{dT}{dt} = -\beta T V_I \tag{1}$$

$$\frac{dI_1}{dt} = \beta T V_I - kI_1 \tag{2}$$

$$\frac{dI_2}{dt} = kI_1 - \delta_x I_2 \tag{3}$$

$$\frac{dV_I}{dt} = p\mu I_2 - cV_I \tag{4}$$

$$\frac{dV_{NI}}{dt} = p(1 - \mu)I_2 - cV_{NI} \tag{5}$$

Based on this model, the basic reproduction number, $R_0$, defined as the number of newly infected cells by one infected cell at the beginning of the infection (*Best et al., 2017*) is, $R_0 = \frac{p\beta T_0 \mu}{c\delta}$. Given the absence of any antiviral effect of hydroxychloroquine against SARS-CoV-2 (*Mitjà et al., 2021*; *Maisonnasse et al., 2020*; *Boulware et al., 2020*), we did not consider any effect of hydroxychloroquine in the model.

## Assumptions on parameter values

Some parameters were fixed to ensure identifiability. The clearance rate $c$ was fixed at $10d^{-1}$ and the eclipse phase $k$ to $4d^{-1}$ based on previous work (*Néant et al., 2021*; *Gonçalves et al., 2020*; *Gonçalves et al., 2021*). The proportion of infectious virus $\mu$ was assumed constant over time and was fixed to $10^{-4}$ as observed in animal model (*Gonçalves et al., 2021*). The initial number of target cells, $T_0$, was fixed to $T_0 = 1.33 \times 10^5 cells.mL^{-1}$ (more details in *Néant et al., 2021*). We assumed that at the moment of infection there was exactly one productively infected cell in the upper respiratory tract. Hence, at $t = t_{inf}, T = T_0; I_1 = 0; I_2 = \frac{1}{30}; V_I = 0$ and $V_{NI} = 0$.

We assumed that the incubation period was lognormally distributed around 5 days before symptoms onset with a standard deviation of 0.125 days, corresponding to 90% of individuals having an incubation period varying between 4 and 6 days (*Jones et al., 2021*; *Lauer et al., 2020*).

## Statistical model for viral kinetics

Parameter estimations were performed using non-linear mixed-effect model. The structural model used to describe the observed $\log_{10}$ viral load is $y_{i,j} = \log_{10} V(t_{i,j}, \Psi_i^V) + e_{i,j}$, where $y_{i,j}$ is the $j^{th}$ observation of index $i$ at time $t_{i,j}$ with $i\epsilon 1, \ldots, N$ and $j\epsilon 1, \ldots, n_i$ with $N$ the number of index and $n_i$ is the number of observations for index $i$. $V(t_{i,j}, \Psi_i^V)$ is the function describing the total viral load dynamics $V_I(t_{i,j}) + V_{NI}(t_{i,j})$ predicted by the model at time $t_{i,j}$. The vector of viral kinetic parameters for index $i$ is noted $\Psi_i^V$ and $e_{i,j}$ is the additive residual Gaussian error term of constant standard deviation $\sigma$. The vector of individual parameters depends on a fixed effects vector and on an individual random effects vector, which follows a normal centred distribution with a diagonal variance-covariance matrix $\Omega$. To ensure positivity, the individual parameters follow a lognormal distribution.

## Probability of transmission

We noted $x_i^c$ the outcome of the $c^{th}$ contact of index case $i$ (i.e. $x_i^c = 1$ if the contact resulted in transmission and 0 otherwise) and $c\epsilon 1, \ldots, C_i$, with $C_i$ the number of contacts of index $i$. The probability of transmission depends on the time of contact $t_i^c$, the nature of contact, namely household $(h_i^c = 1)$ or not $(h_i^c = 0)$, and the vector of individual parameters $\Psi_i$, which contains the viral parameters $\Psi_i^V$ and individual transmission parameters $\beta_i$. Three models of transmission were tested (M1-M3), described as follows:

### Model M1

No effect of viral load.

$$\text{logit } P(x_i^c = 1 | t_i^c, \Psi_i, h_i^c) = \alpha + \beta_i$$

where: $\beta_i = (\gamma_1 h_i^c + \gamma_0 (1 - h_i^c)) \times \exp(b_i)$ with $\gamma_1$ (resp. $\gamma_0$) the effect of household contact (resp. non-household) on the probability of transmission, and $b_i$ is an individual random effect assumed to follow a Gaussian distribution of variance $\omega_\beta^2$. The baseline probability of transmission was fixed to 5% ($\alpha = -2.94$).

### Model M2

Logit-linear effect of viral load.

$$\text{logit } P(x_i^c = 1 | t_i^c, \Psi_i, h_i^c) = \begin{cases} \alpha \text{ if } \log_{10} V(t_i^c, \Psi_i^V) \leq 6 \\ \alpha + \beta_i \times (\log_{10} V(t_i^c, \Psi_i^V) - 6) \text{ if } \log_{10} V(t_i^c, \Psi_i^V) > 6 \end{cases}$$

where: $\beta_i = (\gamma_1 h_i^c + \gamma_0 (1 - h_i^c)) \times \exp(b_i)$ with $\gamma_1$ (resp. $\gamma_0$) the effect of viral load on the probability of transmission in household contact (resp. non-household), and $b_i$ a Gaussian individual random effect with variance $\omega_\beta^2$. The baseline probability of transmission was fixed to 5% ($\alpha = -2.94$) for viral load lower than 6 $\log_{10}$ copies per mL, which corresponds to the threshold for viral culture (*Néant et al., 2021*; *Ke et al., 2020*) (see *Supplementary file 2* for additional scenarios with different threshold values).

### Model M3

Log-linear effect of viral load.

$$\log P(x_i^c = 1 | t_i^c, \Psi_i, h_i^c) = \begin{cases} \alpha \text{ if } \log_{10} V(t_i^c, \Psi_i^V) \leq 6 \\ \alpha + \beta_i \times (\log_{10} V(t_i^c, \Psi_i^V) - 6) \text{ if } \log_{10} V(t_i^c, \Psi_i^V) > 6 \end{cases}$$

where: $\beta_i = (\gamma_1 h_i^c + \gamma_0 (1 - h_i^c)) \times \exp(b_i)$ with $\gamma_1$ (resp. $\gamma_0$) the effect of viral load on the probability of transmission in household contact (resp. non-household), and $b_i$ a Gaussian individual random effect with variance $\omega_\beta^2$. The baseline probability of transmission was fixed to 5% ($\alpha = -2.99$) and the probability was bounded to 1.

## Parameter estimation

For each model, we estimated simultaneously the vector of individual parameter $\Psi_i$, which depends on both the parameters of the viral kinetic model $(R_0, \delta, p, \omega_{R_0}, \omega_\delta, \omega_p)$ and the parameters of the transmission model $(\beta, \omega_\beta)$. The model providing the lowest BIC was retained. All parameters were estimated by computing the maximum-likelihood estimator using the stochastic approximation expectation-maximization (SAEM) algorithm implemented in Monolix Software 2020R1 (http://www.lixoft.eu/) (*Comets et al., 2017*; *Delyon et al., 1999*; *Monolix version 2020R1, 2019*).

## Simulations settings

We provided prediction intervals for viral load and transmission probability over time, depending on the nature of contact, namely household $(h = 1)$ or not $(h = 0)$. We sampled $M = 1,000$ individual from the estimated population distribution and we calculated the predicted viral load $V(t, \Psi_m^V)$ and the predicted transmission probability according to the type of contact $P(x_m | t, \Psi_m, h)$ for all $M$ individuals. We derived the mean transmission probability over the $M$ simulated individuals at all times, as well as the 90% inter quantile range to provide prediction intervals.

All simulations were performed using the Simulx package on R.3.6.0.

## Calculating the average probability of transmission

Using our model, we also aimed to visualise the impact of a therapeutic intervention or a virus mutation on the probability of transmission. To this purpose, we defined several scenarios of simulation by modifying the corresponding parameters in the viral dynamic model. First, we increased the viral production parameter, $p$, by a factor of 2 to 100 corresponding to observed increases of 1-7 $C_t$ value for different variants (*Teyssou et al., 2021*; *Roquebert et al., 2021*; *Kidd et al., 2021*). Second, we decreased the production parameters $p$ by a factor of 2, 4, 8, and 16 as well (*Liu et al., 2021*) to emulate the impact of vaccination (*Levine-Tiefenbrun et al., 2021*; *McEllistrem et al., 2021*; *Figure 4—figure supplement 1*).

We used as a metrics of the effect of variants the average probability of transmission during the contact period, defined as

$$\overline{P_h} = \int_{m,t} P(x_m = 1 | t, \Psi_m, h) g(t) d\Psi_m dt$$

where $P(x_m = 1 | t, \Psi_m, h)$ is the probability of infection after a high-risk contact occurring at time $t$ given the parameters of individual $m$. We considered two possible distributions of contacts $g(t)$, (i) a constant function during the first five days following infection, followed by a decreasing function afterwards, reflecting the time-decreasing likelihood of contacts due to detection and/or symptom onset; (ii) a constant function during the first 20 days following infection (e.g. uniform distribution of the contact).

## Generation interval

As a means to validate the model predictions, we also calculated the generation interval, defined as the time between the infection of the index and the infection of the contact. Given the difficulty to account for random effects, the generation time was calculated by simulations as follows.

We first sampled a vector of individual parameter $\Psi_m$ in the simulated population distribution. We then sampled a time of contact $t_c$ in the contact distribution. Finally, the contact outcome (i.e. infection or not) was obtained by drawing in the binomial distribution of parameter $P(x_m = 1 | t_m = t_c, \Psi_m, h_m)$. We repeated these steps 500,000 times to obtain the distribution of the generation time.

## Acknowledgements

The study has received financial support from the National Research Agency (ANR) through the ANR-Flash call for COVID-19 (Grant ANR-20-COVI-0018) and the Bill and Melinda Gates Foundation under Grant Agreement INV-017335. The original trial was funded by a crowdfunding campaign YoMeCorono (https://www.yomecorono.com/), and Generalitat de Catalunya with support for laboratory equipment from Foundation Dormeur. The sponsors had no role in the conduct of the trial,

the analysis, or the decision to submit the manuscript for publication. The trial protocol and subsequent amendments were approved by the institutional review board at Hospital Germans Trias i Pujol and the Spanish Agency of Medicines and Medical Devices. All the participants provided written informed consent. (https://www.nejm.org/doi/10.1056/NEJMoa2021801).We thank Samuel Alizon, Xavier Duval and Xavier de Lamballerie for helpful discussions.

## Additional information

### Funding

| Funder | Grant reference number | Author |
| --- | --- | --- |
| Bill and Melinda Gates Foundation | INV-017335 | Jeremie Guedj |
| French National Research Agency | ANR-20-COVI-0018 | Jeremie Guedj |
| European Research Council | ERC Starting Grant under the European Union's Horizon 2020 research and innovation programme | Oriol Mitjà |
| YoMeCorono | Crowdfunding campaign | Oriol Mitjà |
| Generalitat de Catalunya | | Oriol Mitjà |

The funders had no role in study design, data collection and interpretation, or the decision to submit the work for publication.

### Author contributions

Aurélien Marc, Marion Kerioui, Modelling, Formal analysis, Methodology, Writing - reviewing and editing; François Blanquart, Formal analysis, Writing - reviewing and editing; Julie Bertrand, Formal analysis, Writing - review and editing; Oriol Mitjà, Marc Corbacho-Monné, Resources, Writing - review and editing; Michael Marks, Formal analysis, Resources, Writing - review and editing; Jeremie Guedj, Conceptualization, Supervision, Funding acquisition, Methodology, Project administration, Writing - reviewing and editing

### Author ORCIDs

Aurélien Marc https://orcid.org/0000-0002-6936-5388
Jeremie Guedj https://orcid.org/0000-0002-5534-5482

### Ethics

Clinical trial registration NCT04304053.
Human subjects: The trial was supported by the crowd funding campaign YoMeCorono (https://www.yomecorono. com/), Generalitat de Catalunya, Zurich Seguros, Synlab Diagnósticos, Laboratorios Rubió, and Laboratorios Gebro Pharma. Laboratorios Rubiódonated and supplied the hydroxychloroquine (Dolquine). The sponsors had no role in the conduct of the trial, the analysis, or the decision to submit the manuscript for publication. The trial protocol and subsequent amendments were approved by the institutional review board at Hospital Germans Trias i Pujol and the Spanish Agency of Medicines and Medical Devices. All the participants provided written informed consent. (https://www.nejm.org/doi/10.1056/NEJMoa2021801).

### Decision letter and Author response
Decision letter https://doi.org/10.7554/eLife.69302.sa1
Author response https://doi.org/10.7554/eLife.69302.sa2

## Additional files

### Supplementary files

• Source data 1. Viral load of index cases and list of their high-risk contacts.

• Supplementary file 1. Parameter estimates of models with a fixed incubation period ranging from 4 to 7 days.

• Supplementary file 2. Parameters estimates of models with different threshold values below which the transmission is set to 5%.

• Supplementary file 3. Parameter estimates of the Model M2 (Left) and the same model without variability in transmission (Right).

• Supplementary file 4. Proportion of contacts wearing masks in each category (Left). Secondary attack rate calculated for each category (Right).

• Transparent reporting form

### Data availability

All data used in this study have been included in the supporting files. The dataset can be found in Marks et al, *The Lancet*, 2021.

The following datasets were generated:

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
