## [Decision Letter]

**Acceptance summary:**

This study carefully evaluates the relationship between viral load and infectiousness by coupling data on viral load with information from epidemiological contact tracing. The quantification of household and non-household transmission as a function of viral load is an important advance in SARS-CoV-2 epidemiology with implications for other respiratory pathogens.

**Decision letter after peer review:**

Thank you for submitting your article "Quantifying the relationship between SARS-CoV-2 viral load and infectiousness" for consideration by *eLife*. Your article has been reviewed by 2 peer reviewers, and the evaluation has been overseen by a Reviewing Editor and Jos van der Meer as the Senior Editor. The reviewers have opted to remain anonymous.

Essential revisions:

The reviewers and the editors find these results interesting and a potentially valuable contribution, assuming they can be shown to be robust to several modeling assumptions and limitations of the data, and these caveats more directly conveyed in the text. The following major suggestions emerged from the reviews and consultation session:

1. Data on contacts and viral loads are particularly weak during the time during which most transmission takes place, i.e., during the presymptomatic and early symptomatic period. How do the results change with other methods of extrapolating peak viral load, including allowing variation in the incubation period? Are the results impacted if participants are assumed to report contacts most faithfully around the day of symptom onset?

2. Mask use was recorded and surely affects the "effective" viral load for infectiousness. Does accounting for it change the results?

3. Reviewer 2 highlighted ambiguity about the importance of variation in infectiousness between individuals v. over time. Do the data support real differences between people (after accounting for mask use and contacts)?

4. Both reviewers also expressed skepticism about the strength of evidence underlying VOC viral loads.

I'd like to clarify that the main suggestion here is for a more careful analysis. If the results change (assuming key parameters remain identifiable), this paper can still make a useful contribution.

Please see the two reviews for more detailed suggestions.

*Reviewer #1:*

The study of Marc et al., evaluated the relationship between viral load and infectiousness using a set of data containing both viral load and epidemiological contact tracing information collected from infected individuals during the first wave of SARS-CoV-2 outbreak in Spain (i.e. March-April, 2020). A subset of individuals where both longitudinal viral load measurements and high-risk of contacts were reported. The authors first fit a viral dynamic model, i.e. a type of model describing viral infection process in infected individuals, to longitudinal viral load data. This model gives predictions of viral load trajectories over time including the viral load at the time of the high-risk contacts. Then regression models were used to assess how the probability of transmission changes with viral loads. The authors found that the risk of transmission differs between house-hold contacts and non-house hold contacts; and for both types of contacts, the risk increases with increases in Log viral load. The authors also use the model to evaluate the transmissibility of the variant B117 assuming B117 causes a 2-4 fold higher viral load.

This is an interesting and novel study that addresses an important question: how viral load is related to transmission. An accurate understanding of this question will help to understand how the transmission patterns at the epidemiological level are driven by viral progression at the individual level and to predict impacts of interventions that reduce viral loads on transmission.

There are several limitations in the datasets and model predictions (as the authors have already rightly pointed out). Some of these limitations may potentially impact the conclusion. Rigorous robustness analyses need to be performed to test whether the conclusions are sensitive to the assumptions in the model or limitations in the datasets.

First, limitation in the contact tracing data. Supplementary Figure 2 shows that a high level of contacts occurred at the time of symptom onset. This is a strong signal of certain biases in the survey data. Usually, one would expect that either contacts are roughly uniformly distributed (in the case of mildly symptomatic cases) or the number of contacts is higher before symptom onset and lower after symptom onset (for severely symptomatic cases). The data shown in supplementary Figure 2 clearly do not follow any of these patterns, i.e. it shows that the number of contacts suddenly become high on the day of symptom onset and decreased afterwards. One potential explanation of this observation is that when people start to have symptoms, they are more aware of the contacts they made on the day of symptom onset and a few days before and afterwards. This could influence the results in the study.

Second, limitation in the predicted viral loads at the time of high-risk contacts. It seems that there are 3 viral load measurements per infected individual in general, and these measurements were made approximately 4 days post symptom onset. In contrast, the transmission events and contacts were mostly occurred before or at symptom onset, i.e. several days before the first viral load data is available. Therefore, the viral load at the time of contacts were back extrapolated from data. Statistically speaking, back extrapolation using 3 data points would work well if the underlying function is a straight line; however, viral load kinetics seem to be more complicated. It is not clear to me whether this back extrapolation is accurate. Furthermore, as the authors pointed out, the model assumes a fixed incubation period of 5 days. However, the incubation period ranges widely between 1-14 days as most frequently quoted. All these uncertainties need to be formally addressed (by testing different scenarios) before we can be sure that the predicted viral loads at the time of contacts are accurate and thus the inferred relationship are reliable.

Third, limitation in understanding of the viral load time course of the B117 strain. The authors' analysis assumes that the strain has a 2-4 fold higher viral load than WT (or non-B117) strain. I am not very convinced by these assumptions. How the viral load trajectories of B117 differ from other strains are not well established due to lack of longitudinal data. These estimates seem to be made mostly based on cross sectional studies, where the viral loads measured in cross sectional studies can be influenced by many factors (for example, the stage of the epidemic) in addition to true differences in viral load trajectories. In addition, transmission potential is influenced by multiple factors (in addition to viral load), such as the resulting severity of infection, the ability to initiate an infection etc. Therefore, I think the predictions concerning the transmission potential of this strain is premature.

To address the first limitation, this potential bias in the contact tracing data shall be accounted before the data is used to calculate how the risk of transmission is related to viral load, unless the authors provide a convincing explanation/argument for the observed pattern.

To address the second limitation, one suggestion I have is to run a simulation study to test accuracy of the predicted viral load at symptom onset (when the assumptions in the model are relaxed). For example, in the simulations, incubation period ranges between 1-14 days, and synthetic data are collected after day 4 post symptom onset (with certain measurement noise). One can fit the model to the synthetic data assuming a 5-day incubation period and test how accurate the inferred viral load at the time of the contacts compared to the viral load in the simulations that generated the data.

*Reviewer #2:*

Using data from a cluster-randomised trial of the use of hydroxychloroquine as prophylaxis in the prevention of SARS-CoV-2 infection among exposed close contacts [Mitja et al., NEJM 2021] and continuing their investigation on factors affecting transmissibility including viral shedding (load) of the index cases [Marks et al., Lancet Inf Dis 2021], in this manuscript, Marc et al. attempted to define the quantitative relationship between viral shedding of the index cases and the transmission probability to close contacts using joint models. It was done by first modelling the viral kinetics of the index using a within-host (target cell-limited) model, and then using probabilistic models with data on observed viral load and secondary transmission events to estimate the parameters of viral dynamics and transmission probability. Their results suggested (1) higher viral load was associated with higher transmission probability (but was not a linear relationship); (2) the effect of viral load on transmission probability was more prominent in households than other settings such as healthcare settings or nursing homes, with as much as 37% when viral load was >10 log10 copies/mL; (3) transmission probability peaked at symptom onset of the index; and (4) based on viral shedding data of different variants, one may estimate the transmission probability of emerging variants such as B1.1.7. The authors suggested/ concluded that such analytic approach could help inform the effects of virus evolution or vaccination on transmission probability.

Authors state in their introduction that "…the role of the viral load as a driver of infectiousness has been suspected but not formally established". However, this statement is incorrect as the link between viral load and infectiousness has been known for more than a year for example as reported by He et al., (Nat Med 2020), among others. Authors reproduced this observation and therefore this work does not appear to make a substantial contribution to knowledge. On the other hand, this manuscript is well-written, using a unique dataset, earlier publications provided detailed description of the data used, the analytical approach used in this manuscript was clearly described with data used made available allowing reproducibility, and limitations adequately acknowledged. Unfortunately, the limitations described below, some of which also acknowledged by the authors, would suggest that the reliability of the identified quantitative relationship between viral load and transmission probability in this manuscript is unclear.

The manuscript seems has confused over two separate issues. First, viral loads in the respiratory tract are known to peak at around the time of symptom onset and then decline, consistent with the overall trajectory in contagiousness. Second, there is variability in viral loads between individuals, for example suggested by Chen et al., (*eLife* 2021, https://elifesciences.org/articles/65774), but it is not clear whether the individuals with higher viral loads are more contagious. Authors do seem to allow for person-to-person variability in their analysis, but it is not clear to this reviewer whether the person-to-person variability is necessary to explain contagiousness. What I believe authors should have done is fitted a model with temporal variability in shedding, and compared to this a model with temporal and person-to-person variability, to determine whether the person-to-person variability is correlated with transmission. In other words, do people with higher peak viral load (or longer duration in shedding) have higher contagiousness?

Measuring virus in respiratory swabs only and use it as a proxy of viral shedding/ infectiousness of the host as a whole is also unlikely to tell the full story. Prior research such as those by Milton et al., (PNAS 2018) and Leung et al., (Nat Med 2020) have shown, depending on the respiratory virus studied, there is a possibility of relatively weak correlation between viral loads in respiratory swabs versus in exhaled aerosols. Depending on the relative importance of different modes of transmission, the different viral shedding at different 'sites' may have implications on the relationship between viral shedding and host's contagiousness.

Therefore, although authors noted on page 8 that "several other factors are involved in transmission, besides viral load", I would posit one factor would be individual variation in viral load that does not seem to have been taken into account, and a second factor would be the difference between viral load in exhaled breath versus viral load in respiratory swab. Both of these factors would actually count as "viral load" factors, rather than "factors … besides viral load".

In addition, although in Marks et al., (Lancet Inf Dis 2021) the use of facemasks by contacts was not identified as a significant factor associated with transmission, the effect of the use of facemasks by index was not assessed despite the data was collected as described in the study protocol from Mitja et al., (NEJM 2021). This is likely an important factor on "effective" viral shedding of the index [Leung et al., Nat Med 2020] which was not accounted for in the present analyses, and based on the data that around 60% of contacts reported routine use of masks [Mitja et al., NEJM 2021], it was likely that a substantial proportion of index would have worn masks too.

Authors go on to extrapolate to VOCs, and note "In the context of variants of concern, that are likely associated with larger viral loads, it becomes even more critical to delineate the contribution of viral shedding from other suspected factors associated with an increased transmission.", although the relationship between different strains and respiratory swab viral load is still unclear, and no available data so far on viral load in exhaled aerosols for different virus strains.

Additional limitations included (1) the unknown timing of the effective contact between the index and exposed contact leading to transmission, due to the unrecognised risk of pre-symptomatic transmission at the time the study was conducted so that most (household) contacts were reported to have happened on the day of symptom onset, and the inability to single-out the contact episode among several repeated contact episodes that actually led to transmission; and (2) the difficult in identifying the viral load during the presymptomatic phase of the index due to lack of data. Overall, I agree with the authors that to identify the quantitative relationship between viral shedding and transmissibility probability for SARS-CoV-2 in the present study (or any future studies) is challenging due to the difficulty in collecting viral shedding data during the presymptomatic transmission phase of SARS-CoV-2.

[Editors' note: further revisions were suggested prior to acceptance, as described below.]

Thank you for submitting your article "Quantifying the relationship between SARS-CoV-2 viral load and infectiousness" for consideration by *eLife*. Your article has been reviewed by 2 peer reviewers, and the evaluation has been overseen by a Reviewing Editor and Jos Van der Meer as the Senior Editor. The reviewers have opted to remain anonymous.

Essential revisions:

This study evaluates the relationship between viral load and infectiousness and is of potential interest to infectious disease modelers and policy makers. The work reaches similar conclusions to other recent studies, although significant uncertainties remain.

The revisions have greatly improved the manuscript, and the reviewers and I ask only that small changes be made for clarity, to help future readers.

1) As suggested by reviewer 2, please mention the lack of information on face mask usage in index cases and the impact this might have on the results.

2) As suggested by reviewer 1 (point 2), please reconsider whether the data have sufficient power to demonstrate saturation and revise accordingly.

3) Please also consider the conflicting evidence on B.1.1.7 viral loads (reviewer 1, point 3).

*Reviewer #1:*

I would like to thank the authors for their efforts to address my comments and concerns. The additional analyses are sufficient and rigorous enough. I still have some concerns with respect to how the results of the study is interpreted and discussed. I would like to recommend publication if these points below are sufficiently addressed.

1. In the revised model fitting, the authors assumed a log-normal distribution for the incubation period (instead of a fixed number). This is a MUCH MORE realistic assumption. However, I still think there are large uncertainties in predicting the viral load at the time of transmission event, especially given that only 3 data points taken on days after transmission events are available. For example, in Figure 1, it seems that the model predicts that the peak viral load occurs in most individuals and the peak viral load is on a back extrapolation of a line from the three data points. It is well known viral load data are very noisy. This extrapolation is unlikely to be very accurate. Although this limitation is partially discussed in lines 241 and 243, I feel this is the uncertainties in predicting pre-peak and peak viral load (where transmission events occurred) that shall be discussed more thoroughly.

2. In line 192-193, it is stated: 'Unlike what has been suggested by theoretical models, the probability of transmission increased continuously with viral load and no saturation effects were visible at high viral loads.' I do not find strong evidence in the manuscript to rule out the possibility that transmission saturates with high viral loads. The saturation effect was not formally tested, because none of the 3 models in the manuscript include the saturation effect. The similar BIC values in Table 1 seem to suggest that this dataset may not be sufficient to test whether a saturation effect exists.

3. For the assumption of increased peak viral load for B.1.1.7, I agree with the authors several cross-sectional studies indicate this VOC had high viral loads. However, as I mentioned in my original review, these studies (including the Jones et al., Science study) are mostly from clinical studies where individuals are enrolled days after symptom onset. The type of dataset is not well reliable in predicting peak viral loads of an infection (related to point 1). On the contrary, a recent longitudinal study shows that there is no difference in peak viral loads (most relevant for transmission) between the wild-type and the B.1.1.7 strain (Ke et al., medRxiv; DOI: 10.1101/2021.07.12.21260208). Infectiousness inferred from cell culture data suggests no difference between the wild-type and B.1.1.7 either. Therefore, I do not think there exists a consensus as why B.1.1.7 is more transmissible -factors other than viral load may be important. Having said that, I agree with the authors that the analysis is very useful for VOCs in general, given that some VOCs may have a high viral load as a transmission advantage. Therefore, I feel that the analysis is very valuable, but the conflict findings of B.1.1.7 viral loads shall be fully acknowledged. Currently, the manuscript seems to indicate that it is certain that B.1.1.7 gains transmission advantage through higher viral loads.

4. One complication in the prediction of increased transmissibility of VOCs is that the overall transmission is influenced by both infectiousness (arising from viral loads) and pattern of contacts. The analysis in the manuscript implicitly assumes the contact patterns are the same across these different groups whereas in reality this may not be true. For example, some VOCs may cause more severe infections whereas vaccinated individuals will have milder infections and thus less changes in the number of contacts. The assumption is ok (without data on contact patterns); but it is better to state this assumption clearly in the abstract and the discussed in the Discussion, so that the uncertainties/assumptions are transparent to the general readers.

5. Typo: 'different' is repeated in the first sentence in the caption of Figure 4.

---

## [Author Response]

Essential revisions:1. Data on contacts and viral loads are particularly weak during the time during which most transmission takes place, i.e., during the presymptomatic and early symptomatic period. How do the results change with other methods of extrapolating peak viral load, including allowing variation in the incubation period? Are the results impacted if participants are assumed to report contacts most faithfully around the day of symptom onset?

Thank you for this comment. It is correct that we do not have viral load data during the presymptomatic phase where most contacts occurred, as pointed out in Figure 1 —figure supplement 2.

Per your comment, we have relaxed our assumption of a fixed and similar incubation duration. We now use in all our models a lognormal distribution with a mean value of 5 days and a standard deviation of 0.125 days, representing the fact that 90% of the incubation times are between 4 and 6 days^1,2^. Additionally, we have extended in the supplementary materials our sensitivity analyses by assuming a fixed incubation period and tested values ranging from 4 days to 7 days. All models yielded similar results, showing a significant effect of the viral load on the transmission for both non-household and household contacts (Supplementary File 1).

We have modified the description of the model accordingly in the methods and updated all our results in the revised version of the manuscript.

2. Mask use was recorded and surely affects the "effective" viral load for infectiousness. Does accounting for it change the results?

Unfortunately, the study did not contain detailed information on the mask use by the index cases. As reported in Marks et al.^3^ the information collected was the routine use of face mask by contacts when in close proximity to the index case, and this was not found associated with transmission^3^. This may be due to several reasons, including the poor reporting of this information, that was missing for 35% of contacts (Supplementary Table 3 and more discussion in Marks et al.,).

Given these limitations, we have decided not to include mask use in our model but we now clearly add this as a limitation in the discussion:

“Beside viral load, several factors are associated with a transmission event. One important one is face masking, for both the index and the contact. In the original analysis of Marks et al.^3^, the use of face mask by contacts was not found associated with a decreased viral load, but this probably reflects the lack of more detailed data on the type of mask, the use of other personal protective equipment and infection control practices. It is also important to recall that face masking was poorly reported and was missing in about 35% of contacts, limiting statistical power (Supplementary file 4). Collecting this information in future studies should probably contribute to a reduction in the variance of the random effect parameter associated with transmission (ωβ).”

3. Reviewer 2 highlighted ambiguity about the importance of variation in infectiousness between individuals v. over time. Do the data support real differences between people (after accounting for mask use and contacts)?

It is important to realize that our model allows variability on both the viral dynamics and the individual risk of transmission after adjustment on viral load. This is accounted by the parameter βi in all 3 models of transmission tested. This allows, in other words, two index cases having similar viral load to have nonetheless different probability of transmission, that could be due to many individual or behavioral factors not represented in the model.

Per your comment, we have also tested a model without variability in transmission, and this yields to similar estimates of the viral load parameters but increased parameter linking viral load and infectiousness (Supplementary file 2). The effect of the viral load is still significant in both household and non-household contacts (wald test p-value <0.01) but the loglikelihood is increased by more than 10 points, leading to model rejection over a model with variability.

We clarified this aspect in the Results:

“The model considers two levels of individual variability, one on the viral load dynamics^4^ (as measured by the standard deviation of the associated random effects, ωR0,ωδ and ωp), and another one on the probability of transmission (with a standard deviation ωβ). Of note, ωβ was equal to 85%, indicating that several other factors are involved in the transmission probability, even after adjustment for viral load levels (see Supplementary file 3 for the results obtained with a model assuming a similar value for β in all individuals).” 4. Both reviewers also expressed skepticism about the strength of evidence underlying VOC viral loads.

We respectfully disagree with the reviewer on that aspect. Data have accumulated on the effects of VOC on viral load. Most studies indeed rely on large cross sectional studies ^5–8^ but our group has also been involved in studies with longitudinal follow-up^9,10^ ; our results confirmed that B1.1.7 is associated with a higher viral load, with an estimate of a 2 to 4-fold higher viral load (corresponding to a difference of 1 to 2 Ct values compared to the historical variant). In the recent study from Christian Drosten group, an even higher estimate was found, with a 1 log_10_ higher viral load in individuals infected with B1.1.7 virus compared to the historical variant^2,8,11^. The estimates for the P1 and B1.1.351 are much less well established, with values ranging from a 2-fold^8^ to a 10-fold^11^ increase. We do not mention recent reports of a 1000-fold increase caused by δ virus, that have not been yet confirmed by other studies^12^.

Given the rapidly changing landscape of VOC and the uncertainty on the magnitude of current and future VOC, we provided predictions with a large range of scenarios, that could be relevant with other emerging VOC.

I'd like to clarify that the main suggestion here is for a more careful analysis. If the results change (assuming key parameters remain identifiable), this paper can still make a useful contribution.Please see the two reviews for more detailed suggestions.Reviewer #1:[…]To address the first limitation, this potential bias in the contact tracing data shall be accounted before the data is used to calculate how the risk of transmission is related to viral load, unless the authors provide a convincing explanation/argument for the observed pattern.

We agree that contact tracing does not prevent from recollection biases, and this could explain the over representation of contacts at symptom onset shown in Figure 1 —figure supplement 2. As recollection bias equally affects contacts that have led to an infection from those that did not lead to an infection, this creates a mechanisms of data missingness called “missing at random”, which does not bias the parameter estimation for the relationship between viral load and transmission.

However, it is correct that a potential over-representation of the contacts at symptom onset in the original data set may create bias in the prediction of the effects of VOC on transmission, that depends on the assumption made for the distribution of contacts.

Following reviewer’s comments, we have redone all our simulations to consider two more realistic scenarios for the contact distribution: (i) a constant function during the first five days following infection, followed by a decreasing function afterwards, reflecting the time-decreasing likelihood of contacts due to detection and/or symptom onset. (ii) a constant function during the first 20 days following infection (eg, uniform distribution of the contact). Of note, scenario (i) reflects the fact that contacts are less likely to occur after 5 days, which corresponds to the typical duration of the incubation period. Given the absence of data on the relationship between symptom onset and contacts, the distribution of symptom onset and the distribution of contacts were considered as independent.

All results have been modified accordingly.

To address the second limitation, one suggestion I have is to run a simulation study to test accuracy of the predicted viral load at symptom onset (when the assumptions in the model are relaxed). For example, in the simulations, incubation period ranges between 1-14 days, and synthetic data are collected after day 4 post symptom onset (with certain measurement noise). One can fit the model to the synthetic data assuming a 5-day incubation period and test how accurate the inferred viral load at the time of the contacts compared to the viral load in the simulations that generated the data.

We thank you for this important comment. We have now relaxed the assumption of a fixed and similar incubation period and we now assume a log-normal distribution of the incubation period with a mean value of 5 days and a standard deviation for the random effect of 0.125 days, to ensure a 90% probability that the incubation time is between 4 and 6 days^1,2^. To ensure the consistency of our results, we also provided in the supplementary the results of models assuming a fixed incubation period ranging from 4 days to 7 days. All models yielded similar results, showing a significant effect of the viral load on the transmission for both non-household and household contacts (Supplementary file 1).

Third, limitation in understanding of the viral load time course of the B117 strain. The authors' analysis assumes that the strain has a 2-4-fold higher viral load than WT (or non-B117) strain. I am not very convinced by these assumptions. How the viral load trajectories of B117 differ from other strains are not well established due to lack of longitudinal data. These estimates seem to be made mostly based on cross sectional studies, where the viral loads measured in cross sectional studies can be influenced by many factors (for example, the stage of the epidemic) in addition to true differences in viral load trajectories. In addition, transmission potential is influenced by multiple factors (in addition to viral load), such as the resulting severity of infection, the ability to initiate an infection etc. Therefore, I think the predictions concerning the transmission potential of this strain is premature.

We respectfully disagree with the reviewer on that aspect. Data have accumulated on the effects of VOC on viral load. Most studies indeed rely on large cross sectional studies ^5–8^ but our group has also been involved in studies with longitudinal follow-up^9,10^ ; our results confirmed that B1.1.7 is associated with a higher viral load, with an estimate of a 2 to 4-fold higher viral load (corresponding to a difference of 1 to 2 Ct values compared to the historical variant). In the recent study from Christian Drosten group, an even higher estimate was found, with a 1 log_10_ higher viral load in individuals infected with B1.1.7 virus compared to the historical variant^2,8,11^. The estimates for the P1 and B1.1.351 are much less well established, with values ranging from a 2-fold^8^ to a 10-fold^11^ increase. We do not mention recent reports of a 1000-fold increase caused by delta virus, that have not been yet confirmed by other studies^12^.

Given the rapidly changing landscape of VOC and the uncertainty on the magnitude of current and future VOC, we provided predictions with a large range of scenarios, that could be relevant with other emerging VOC.

Reviewer #2:[…]Authors state in their introduction that "…the role of the viral load as a driver of infectiousness has been suspected but not formally established". However, this statement is incorrect as the link between viral load and infectiousness has been known for more than a year for example as reported by He et al., (Nat Med 2020), among others.

We have clarified our wording:

“While respiratory droplets and aerosols have been rapidly demonstrated to be a major route of transmission of SARS-CoV-21, the role of the viral load as a driver of infectiousness has been established but not formally quantified.”

Authors reproduced this observation and therefore this work does not appear to make a substantial contribution to knowledge. On the other hand, this manuscript is well-written, using a unique dataset, earlier publications provided detailed description of the data used, the analytical approach used in this manuscript was clearly described with data used made available allowing reproducibility, and limitations adequately acknowledged. Unfortunately, the limitations described below, some of which also acknowledged by the authors, would suggest that the reliability of the identified quantitative relationship between viral load and transmission probability in this manuscript is unclear.The manuscript seems has confused over two separate issues. First, viral loads in the respiratory tract are known to peak at around the time of symptom onset and then decline, consistent with the overall trajectory in contagiousness. Second, there is variability in viral loads between individuals, for example suggested by Chen et al., (eLife 2021, https://elifesciences.org/articles/65774), but it is not clear whether the individuals with higher viral loads are more contagious. Authors do seem to allow for person-to-person variability in their analysis, but it is not clear to this reviewer whether the person-to-person variability is necessary to explain contagiousness. What I believe authors should have done is fitted a model with temporal variability in shedding, and compared to this a model with temporal and person-to-person variability, to determine whether the person-to-person variability is correlated with transmission. In other words, do people with higher peak viral load (or longer duration in shedding) have higher contagiousness?

Thank you for this relevant reference, which has been added.

It is important to realize that our model allows variability on both the viral dynamics and the individual risk of transmission after adjustment on viral load. This is accounted by the parameter β_i_ in all 3 models of transmission tested. This allows, in other words, two index cases having similar viral load to have nonetheless different probability of transmission, that could be due to many individual or behavioral factors not represented in the model.

Per your comment, we have also tested a model without variability in transmission, and this yields to similar estimates of the viral load parameters but increased parameter linking viral load and infectiousness (Supplementary Table 2). The effect of the viral load is still significant in both household and non-household contacts (wald test p-value <0.01) but the loglikelihood is increased by more than 10 points, leading to model rejection over a model with variability.

We clarified this aspect in the Results:

“The model considers two levels of individual variability, one on the viral load dynamics^4^ (as measured by the standard deviation of the associated random effects, ωR0,ωδ and ωp), and another one on the probability of transmission (with a standard deviation ωβ). Of note, ωβ was equal to 85%, indicating that several other factors are involved in the transmission probability, even after adjustment for viral load levels (see Supplementary file 3 for the results obtained with a model assuming a similar value for β in all individuals).”

References

1. Lauer, S. A., Grantz, K. H., Bi, Q., Jones, F. K., Zheng, Q., Meredith, H. R., Azman, A. S., Reich, N. G. and Lessler, J. The Incubation Period of Coronavirus Disease 2019 (COVID-19) From Publicly Reported Confirmed Cases: Estimation and Application. *Ann Intern Med* 172, 577–582 (2020).

2. Jones, T. C. *et al.,* Estimating infectiousness throughout SARS-CoV-2 infection course. *Science* eabi5273 (2021) doi:10.1126/science.abi5273.

3. Marks, M., Millat-Martinez, P., Ouchi, D., Roberts, C. h, Alemany, A., Corbacho-Monné, M., Ubals, M., Tobias, A., Tebé, C., Ballana, E., Bassat, Q., Baro, B., Vall-Mayans, M., G-Beiras, C., Prat, N., Ara, J., Clotet, B. and Mitjà, O. Transmission of COVID-19 in 282 clusters in Catalonia, Spain: a cohort study. *The Lancet Infectious Diseases* 0, (2021).

4. Chen, P. Z., Bobrovitz, N., Premji, Z., Koopmans, M., Fisman, D. N. and Gu, F. X. Heterogeneity in transmissibility and shedding SARS-CoV-2 via droplets and aerosols. *eLife* 10, e65774 (2021).

5. Kidd, M., Richter, A., Best, A., Cumley, N., Mirza, J., Percival, B., Mayhew, M., Megram, O., Ashford, F., White, T., Moles-Garcia, E., Crawford, L., Bosworth, A., Atabani, S. F., Plant, T. and McNally, A. S-variant SARS-CoV-2 lineage B1.1.7 is associated with significantly higher viral loads in samples tested by ThermoFisher TaqPath RT-qPCR. *The Journal of Infectious Diseases* (2021) doi:10.1093/infdis/jiab082.

6. Calistri, P., Amato, L., Puglia, I., Cito, F., Di Giuseppe, A., Danzetta, M. L., Morelli, D., Di Domenico, M., Caporale, M., Scialabba, S., Portanti, O., Curini, V., Perletta, F., Cammà, C., Ancora, M., Savini, G., Migliorati, G., D’Alterio, N. and Lorusso, A. Infection sustained by lineage B.1.1.7 of SARS-CoV-2 is characterised by longer persistence and higher viral RNA loads in nasopharyngeal swabs. *Int J Infect Dis* 105, 753–755 (2021).

7. Roquebert, B., Haim-Boukobza, S., Trombert-Paolantoni, S., Lecorche, E., Verdurme, L., Foulongne, V., Burrel, S., Alizon, S. and Sofonea, M. T. SARS-CoV-2 variants of concern are associated with lower RT-PCR amplification cycles between January and March 2021 in France. *medRxiv* 2021.03.19.21253971 (2021) doi:10.1101/2021.03.19.21253971.

8. Teyssou, E. *et al.,* The 501Y.V2 SARS-CoV-2 variant has an intermediate viral load between the 501Y.V1 and the historical variants in nasopharyngeal samples from newly diagnosed COVID-19 patients. *Journal of Infection* 0, (2021).

9. Elie, B., Lecorche, E., Sofonea, M. T., Trombert-Paolantoni, S., Foulongne, V., Guedj, J., Haim-Boukobza, S., Roquebert, B. and Alizon, S. Inferring SARS-CoV-2 variant within-host kinetics. *medRxiv* 2021.05.26.21257835 (2021) doi:10.1101/2021.05.26.21257835.

10. Cosentino, G., Bernard, M., Ambroise, J., Giannoli, J.-M., Guedj, J., Débarre, F. and Blanquart, F. SARS-CoV-2 viral dynamics in infections with variants of concern in the French community. (2021).

11. Naveca, F. G. *et al.,* COVID-19 in Amazonas, Brazil, was driven by the persistence of endemic lineages and P.1 emergence. *Nature Medicine* 1–9 (2021) doi:10.1038/s41591-021-01378-7.

12. Viral infection and transmission in a large well-traced outbreak caused by the Delta SARS-CoV-2 variant - SARS-CoV-2 coronavirus / nCoV-2019 Genomic Epidemiology. *Virological* https://virological.org/t/viral-infection-and-transmission-in-a-large-well-traced-outbreak-caused-by-the-delta-sars-cov-2-variant/724 (2021).

13. Mollan, K. R. *et al.,* Infectious SARS-CoV-2 Virus in Symptomatic COVID-19 Outpatients: Host, Disease, and Viral Correlates. *medRxiv* 2021.05.28.21258011 (2021) doi:10.1101/2021.05.28.21258011.

14. Néant, N. *et al.,* Modeling SARS-CoV-2 viral kinetics and association with mortality in hospitalized patients from the French COVID cohort. *PNAS* 118, (2021).

15. van Kampen, J. J. A. *et al.,* Duration and key determinants of infectious virus shedding in hospitalized patients with coronavirus disease-2019 (COVID-19). *Nat Commun* 12, 267 (2021).

[Editors' note: further revisions were suggested prior to acceptance, as described below.]

Essential revisions:This study evaluates the relationship between viral load and infectiousness and is of potential interest to infectious disease modelers and policy makers. The work reaches similar conclusions to other recent studies, although significant uncertainties remain.The revisions have greatly improved the manuscript, and the reviewers and I ask only that small changes be made for clarity, to help future readers.1) As suggested by reviewer 2, please mention the lack of information on face mask usage in index cases and the impact this might have on the results.

We have added this information in the discussion:

“The use of face mask by index cases was not collected in the original study. This information might be of a greater importance as it has a far more substantial effect on viral shedding and thus on transmission.”2) As suggested by reviewer 1 (point 2), please reconsider whether the data have sufficient power to demonstrate saturation and revise accordingly.

We have removed this sentence from the discussion.

3) Please also consider the conflicting evidence on B.1.1.7 viral loads (reviewer 1, point 3).

We have acknowledged the limitation due to observational studies:

“We relied on results found in both large-scale epidemiological data and longitudinal evaluation of Ct values ^1,2^, that reported an average increase of the B1.1.7 virus by 1-2 Ct^3–5^, which can be reproduced in our model by assuming that viral production increases by a factor 2-4. […] As raised by one of the reviewers, it is important to recognize that the association between VOC and viral load levels relies on observational studies, with data mostly collected after symptom onset, both factors limiting a formal causation. In fact, another modelling study performed in a small population of frequently sampled individuals diagnosed early in their infection did not find an effect of B1.1.7 on viral kinetics^6^”

Reviewer #1:I would like to thank the authors for their efforts to address my comments and concerns. The additional analyses are sufficient and rigorous enough. I still have some concerns with respect to how the results of the study is interpreted and discussed. I would like to recommend publication if these points below are sufficiently addressed.1. In the revised model fitting, the authors assumed a log-normal distribution for the incubation period (instead of a fixed number). This is a MUCH MORE realistic assumption. However, I still think there are large uncertainties in predicting the viral load at the time of transmission event, especially given that only 3 data points taken on days after transmission events are available. For example, in Figure 1, it seems that the model predicts that the peak viral load occurs in most individuals and the peak viral load is on a back extrapolation of a line from the three data points. It is well known viral load data are very noisy. This extrapolation is unlikely to be very accurate. Although this limitation is partially discussed in lines 241 and 243, I feel this is the uncertainties in predicting pre-peak and peak viral load (where transmission events occurred) that shall be discussed more thoroughly.

We have acknowledged this limitation in the discussion as follows:

“It is also important to note that viral load data in index cases were collected on average 3-4 days after symptom onset, in the declining phase of viral load, several days after most of the contacts had occurred. Although our population parameters were estimated with a reasonable precision (Table 1) it nonetheless brings uncertainty on the predictions of individual trajectories. This limitation is inherent to the nature of SARS-CoV-2, where the peak viral load coincides with symptom onset, making difficult to obtain data during the replicating phase of the virus where individuals are largely asymptomatic.”

2. In line 192-193, it is stated: 'Unlike what has been suggested by theoretical models, the probability of transmission increased continuously with viral load and no saturation effects were visible at high viral loads.' I do not find strong evidence in the manuscript to rule out the possibility that transmission saturates with high viral loads. The saturation effect was not formally tested, because none of the 3 models in the manuscript include the saturation effect. The similar BIC values in Table 1 seem to suggest that this dataset may not be sufficient to test whether a saturation effect exists.

We have removed this part from the discussion.

3. For the assumption of increased peak viral load for B.1.1.7, I agree with the authors several cross-sectional studies indicate this VOC had high viral loads. However, as I mentioned in my original review, these studies (including the Jones et al., Science study) are mostly from clinical studies where individuals are enrolled days after symptom onset. The type of dataset is not well reliable in predicting peak viral loads of an infection (related to point 1). On the contrary, a recent longitudinal study shows that there is no difference in peak viral loads (most relevant for transmission) between the wild-type and the B.1.1.7 strain (Ke et al., medRxiv; DOI: 10.1101/2021.07.12.21260208). Infectiousness inferred from cell culture data suggests no difference between the wild-type and B.1.1.7 either. Therefore, I do not think there exists a consensus as why B.1.1.7 is more transmissible -factors other than viral load may be important. Having said that, I agree with the authors that the analysis is very useful for VOCs in general, given that some VOCs may have a high viral load as a transmission advantage. Therefore, I feel that the analysis is very valuable, but the conflict findings of B.1.1.7 viral loads shall be fully acknowledged. Currently, the manuscript seems to indicate that it is certain that B.1.1.7 gains transmission advantage through higher viral loads.

We have taken into account the point made by reviewer 1 in the discussion:

“We relied on results found in both large-scale epidemiological data and longitudinal evaluation of Ct values ^1,2^, that reported an average increase of the B1.1.7 virus by 1-2 Ct^3–5^, which can be reproduced in our model by assuming that viral production increases by a factor 2-4. […] As raised by one of the reviewers, it is important to recognize that the association between VOC and viral load levels relies on observational studies, with data mostly collected after symptom onset, both factors limiting a formal causation. In fact, another modelling study performed in a small population of frequently sampled individuals diagnosed early in their infection did not find an effect of B1.1.7 on viral kinetics^6^”

4. One complication in the prediction of increased transmissibility of VOCs is that the overall transmission is influenced by both infectiousness (arising from viral loads) and pattern of contacts. The analysis in the manuscript implicitly assumes the contact patterns are the same across these different groups whereas in reality this may not be true. For example, some VOCs may cause more severe infections whereas vaccinated individuals will have milder infections and thus less changes in the number of contacts. The assumption is ok (without data on contact patterns); but it is better to state this assumption clearly in the abstract and the discussed in the Discussion, so that the uncertainties/assumptions are transparent to the general readers.

Reviewer 1 is absolutely right, we assumed the contact pattern to be the same across all VOC, which may not be true. This has been acknowledged in the abstract and in the discussion:

Abstract: “Based on the current knowledge that viral load is increased by 2 to 8-fold with variants of concern and assuming no changes in the pattern of contacts across variants, the model predicts that larger viral load levels could lead to a relative increase in the probability of transmission of 24 to 58% in household contacts, and of 15 to 39% in non-household contacts.”

Discussion: “Also, we assumed the same patterns of contacts in our different scenarios. Although there are no data on these aspects yet, it is possible that larger levels of viral shedding could lead to a more severe infection or, inversely, that lower viral load could produce milder infections, thereby modifying the incubation period and more generally the patterns of contact.”

5. Typo: 'different' is repeated in the first sentence in the caption of Figure 4.

This has been corrected.

References

1. Elie, B., Lecorche, E., Sofonea, M. T., Trombert-Paolantoni, S., Foulongne, V., Guedj, J., Haim-Boukobza, S., Roquebert, B. and Alizon, S. Inferring SARS-CoV-2 variant within-host kinetics. *medRxiv* 2021.05.26.21257835 (2021) doi:10.1101/2021.05.26.21257835.

2. Cosentino, G., Bernard, M., Ambroise, J., Giannoli, J.-M., Guedj, J., Débarre, F. and Blanquart, F. SARS-CoV-2 viral dynamics in infections with variants of concern in the French community. (2021).

3. Teyssou, E. *et al.,* The 501Y.V2 SARS-CoV-2 variant has an intermediate viral load between the 501Y.V1 and the historical variants in nasopharyngeal samples from newly diagnosed COVID-19 patients. *Journal of Infection* 0, (2021).

4. Early analysis of a potential link between viral load and the N501Y mutation in the SARS-COV-2 spike protein | medRxiv. https://www.medrxiv.org/content/10.1101/2021.01.12.20249080v1.

5. Roquebert, B., Haim-Boukobza, S., Trombert-Paolantoni, S., Lecorche, E., Verdurme, L., Foulongne, V., Burrel, S., Alizon, S. and Sofonea, M. T. SARS-CoV-2 variants of concern are associated with lower RT-PCR amplification cycles between January and March 2021 in France. *medRxiv* 2021.03.19.21253971 (2021) doi:10.1101/2021.03.19.21253971.

6. Daily sampling of early SARS-CoV-2 infection reveals substantial heterogeneity in infectiousness | medRxiv. https://www.medrxiv.org/content/10.1101/2021.07.12.21260208v1.